# The Current Trends in Using Nanoparticles, Liposomes, and Exosomes for Semen Cryopreservation

**DOI:** 10.3390/ani10122281

**Published:** 2020-12-03

**Authors:** Islam M. Saadeldin, Wael A. Khalil, Mona G. Alharbi, Seok Hee Lee

**Affiliations:** 1Department of Animal Production, College of Food and Agricultural Sciences, King Saud University, Riyadh 11451, Saudi Arabia; 2Department of Comparative Medicine, King Faisal Specialist Hospital & Research Centre, Riyadh 11211, Saudi Arabia; 3Department of Animal Production, Faculty of Agriculture, Mansoura University, Mansoura 35516, Egypt; w-khalil@mans.edu.eg; 4Department of Biochemistry, College of Sciences, King Saud University, Riyadh 11451, Saudi Arabia; mgalharbi@ksu.edu.sa; 5Center for Reproductive Sciences, Department of Obstetrics and Gynecology, University of California San Francisco, San Francisco, CA 94143, USA

**Keywords:** nanoparticles, liposomes, exosomes, semen, cryopreservation, livestock production

## Abstract

**Simple Summary:**

Long-term preservation of semen is a pivotal step for artificial insemination in most farm animal species, but it is associated with cellular insults at the cell membrane and cytoskeleton level as well as the generation of reactive oxygen species (ROS). We highlight the recent strategies to combat these negative effects through defending against the ROS via antioxidant nanoparticles or through repairing/regenerating the damaged sperm through using liposomes and most recently exosomes derived from the reproductive tract or stem cells.

**Abstract:**

Cryopreservation is an essential tool to preserve sperm cells for zootechnical management and artificial insemination purposes. Cryopreservation is associated with sperm damage via different levels of plasma membrane injury and oxidative stress. Nanoparticles are often used to defend against free radicals and oxidative stress generated through the entire process of cryopreservation. Recently, artificial or natural nanovesicles including liposomes and exosomes, respectively, have shown regenerative capabilities to repair damaged sperm during the freeze–thaw process. Exosomes possess a potential pleiotropic effect because they contain antioxidants, lipids, and other bioactive molecules regulating and repairing spermatozoa. In this review, we highlight the current strategies of using nanoparticles and nanovesicles (liposomes and exosomes) to combat the cryoinjuries associated with semen cryopreservation.

## 1. Introduction

Semen cryopreservation contributes to genetic improvement through artificial insemination, eliminates geographical barriers in artificial insemination (AI) application, and supports the preservation of endangered breeds, thus the conservation of biodiversity. However, the sperm freezing process induces ultrastructural, biochemical, and functional changes of spermatozoa. Especially, spermatozoa membranes and chromatin can be damaged, sperm membrane permeability is increased, and hyper oxidation and formation of reactive oxygen species takes place, affecting fertilizing ability and subsequent early embryonic development [1].

Cryopreservation of mammalian sperm is a complex process affected by several factors for obtaining good quality semen for AI [2], such as type of cryoprotectants or extenders, rates of cooling and thawing, and method of packaging [3,4]. Cryopreservation is associated with damage on the level of the cell membrane, cytoskeleton, DNA, and mitochondria due to the generation of reactive oxygen species (ROS), which affect the entire cellular functions and genome instability [5]. Post-thawing trauma and cellular injury in gametes have been illustrated to affect the cell membrane, organelles, and biochemical perturbation [6]. Sperm cooling and freezing causes membrane phospholipids to accumulate due to van der Waals forces, and transition occurs from liquid crystal phase to gel phase. During thawing, irregular voids occur in the cell membrane that lead to damage to the membrane structure and irregular ion and water leakage both into and out of the cell [7].

In living organisms, generation of ROS, such as hydrogen peroxide (H_2_O_2_), superoxide anions (O_2_^−^), and hydroxyl radicals (OH^−^), may be produced as a result of radiation [8], bio-activation of xenobiotics [9], inflammation [10], cell metabolism [11], decompartmentalization of transition metal ions [12], activities of redox enzymes [13], and deficit in the antioxidant defense [14,15]. Physiologically, free radicals level has a positive impact on sperm cells, including capacitation, hyper-activation, and sperm-oocyte fusion [14]. Therefore, ROS with a physiological limit are required for spermatozoa to attain the fertilizing ability [16], acrosome reaction/acrosomal exocytosis, and sperm motility [17]. However, during semen cryopreservation, the cold shock and the atmospheric oxygen [18,19] increase ROS production and cause an imbalance between free radicals and the antioxidant defense in the semen [20]. Increased ROS production can cause toxic effects in the sperm function [21], in terms of inactivating glycolytic enzymes through acrosomal damage [22], lipid peroxidation (LPO), and reducing sperm fertility [23,24,25]. Notably, LPO is a pathological outcome of several diseases and stress conditions [26]. The LPO process caused by ROS (H_2_O_2_) is detrimental to sperm survivability. As a result of high contents of polyunsaturated fatty acids in the plasma membrane and lack of antioxidant enzymes, mammalian spermatozoa are susceptible to LPO induced damage and loss of sperm functions [27,28]. Increasing ROS generation under oxidative stress (OS) leads to increased sperm plasma membrane failure, damaged spermatozoa [29], reduced sperm cell cytoplasm [30], and finally a marked reduction in viability, the integrity of the sperm membrane, and fertilizing ability and increased damage to sperm DNA [31].

Moreover, the process of freezing has resulted in a significant reduction in GSH content in frozen semen [32,33]. Baghshahi et al. [34] showed that cryopreservation of ram spermatozoa may cause damage to the function and structure of sperm cells, in terms of reduced semen quality and sperm characteristics. This is due to the reduction of the temperature that is associated with the OS, which has been defined as an imbalance between oxidants and cellular antioxidant mechanisms and is induced by the generation of ROS [35]. 

In the last few decades, most of the research work was focused on methods/approaches to improve the freezing efficiency of semen, considered to be a significant issue among reproductive biotechnologists. The approaches employed were mostly based on the protection of spermatozoa against the damaging effects of the freezing procedure, including the use of different extenders, cryoprotectant agents, antioxidants, and nutritional components. Moreover, some reports focused on the repair of the damaged spermatozoa during freezing and thawing. 

There are many potential applications of nanomaterials in farm animal reproduction such as transgenesis and targeted delivery of substances to a sperm cell, antioxidants, antimicrobial properties, and special surface binding ligand functionalization as well as their application in sperm processing and cryopreservation. The antioxidant properties of some nanoparticles (NPs) are among the most promising characteristics for their application in protecting sperm cell functions during cryopreservation [36]. The use of NPs has markedly increased in various fields of animal reproduction including herd fertility issues [36]. Moreover, recent approaches showed the beneficial effects of using liposomes and extracellular vesicles (EVs) including exosomes of different origins to ameliorate the damaging effects of cryopreservation on spermatozoa. In this review, we highlight the recent strategies to defend against or repair the damage that occurs during cryopreservation of semen such as the use of nanoparticles as a defensive approach and nanovesicles including exosomes and liposomes as a repair and defense mechanism for improving the outcomes of semen cryopreservation in different animal species.

## 2. Seminal Plasma, Antioxidants, and Their Effect on Sperm Function

Antioxidants are compounds that scavenge or oppose the actions of ROS [37]. Antioxidants work as chelators or binding proteins, and their three main functions are to suppress the generation of ROS and eliminate ROS that are already present [38].

The antioxidant defense system includes an enzymatic mechanism in seminal plasma and sperm cells such as superoxide dismutase, glutathione reductase, glutathione peroxidase, and catalase. However, the nonenzymatic mechanism includes reduced glutathione (GSH), vitamins (A, C, and E), taurine, and hypotaurine. The rate of LPO in sperm cells is determined by the balance between antioxidative and pro-oxidative mechanisms in the semen [32].

Catalase and superoxide dismutase are antioxidant enzymes, which activate scavenging of ROS. Exposure of spermatozoa, primarily to anaerobic conditions during natural mating, may reduce the number of damaged spermatozoa by ROS. Female oviduct fluids contain substantial taurine levels, as it is an important protective factor of spermatozoa from ROS accumulation [39]. For instance, the catalase enzyme exists in ejaculate for the protection of the spermatozoa through the conversion of H_2_O_2_ into oxygen and water [26]. This prevents the generation of hydroxyl radicals (OH^−^), which are powerful oxidants, by the Fenton reaction [40]. However, bull spermatozoa contain little expression of catalase, which makes them prone to OH^−^ toxicity [41]. Moreover, the concentration of catalase is reduced during semen processing [42]. The addition of antioxidants such as CAT in the buffalo [43], ram [33], boar [44], and bull [45,46,47] semen protected spermatozoa from the damaging effects of ROS and improved motility and membrane integrity during cooling storage. Elevation of the amount of H_2_O_2_ can occur as a result of abnormal sperm with residual cytoplasm or abnormal mid-piece [48]. Equine semen is rich in prostate-derived catalase, and therefore dilution or removing the seminal plasma decreases or adversely affects the scavenging capacity of the ROS [49].

Glutathione (GSH) is a tripeptide that comprises cysteine, glutamate, and glycine ubiquitously expressed in the cells. The cysteine subunit plays a pivotal role in scavenging free radicals. GSH acts as an intracellular defense against OS [50].

Exposure of semen to oxygen and visible light radiation during in vitro fertilization or AI resulted in ROS generation and damaged spermatozoa, reduced motility, and reduced membrane integrity in humans and bovines [51,52,53]. Under these conditions, exogenous addition of catalase, GSH, taurine, superoxide dismutase, and other antioxidants can lead to the maintenance of bovine sperm motility [52]. Supplementation of the whole milk semen extender with hypotaurine or taurine did not improve the motility of bovine spermatozoa in post-thawed semen [54]. In horses, the usage of catalase in extended semen was reported for cooled semen storage [55]. 

As antioxidants reduce the production of free radicals following the freeze–thaw process [56], the application of ROS scavengers is likely to improve sperm function and protect sperm from the deleterious effects of cryopreservation [57,58]. The detrimental effects of cryopreservation could be ameliorated by adding an exogenous source of antioxidants to the freezing medium to reverse OS [32]. This strategy together with other techniques for the removal of defective spermatozoa and cellular debris from semen could be used for gains in the viability of spermatozoa and reducing the necessary spermatozoa to a minimum number per AI dose [20].

## 3. Nanoparticles (NPs)

Several factors affect semen quality and fertilizing ability, including genetic, health, nutrition, season, stresses, and semen cryopreservation [59,60]. Multiple factors lead to poor quality semen [61]. The generation of ROS by nonviable sperm cells in the semen samples impairs sperm function [62]. To obtain good male reproduction, removing unviable or degenerated sperm cells and scavenger ROS from semen samples is important. Recent nanotechnologies reflect new prospects for developing novel and noninvasive techniques for sperm manipulation [63,64,65].

### 3.1. Definition and Characterization of NPs

NPs are molecules with <100 nm diameter and can be applied for different bioapplications including reproductive biology because they have unique physical and chemical properties [60,66]. 

Compared to molecules or bulk solids, there are several differences in the structural properties of the NPs [67]. The key factor of NP activity is the characteristics of their surface, such as size, charge density, and hydrophobicity [68,69]. Manipulation into a nanoform can increase the absorption and bioavailability of the functional ingredients [70]. Particle size can affect or change the properties of the original material [71]. The rapid progress in nanotechnology shows great potential for application in both medical and nutritional sciences because NPs possess unusual and advantageous properties that are different from ordinary or microscale materials in terms of their size and high surface reactivity [72]. NPs have been included in pharmaceuticals to increase the bioavailability of drugs and to target particular tissues/organs [73]. Moreover, NPs show increased cellular uptake, binding properties, and reactivity. Furthermore, the antioxidant properties of NPs recently contributed to optimizing the cryopreservation protocols [74].

Small sizes of nanoparticles have shown better integration possibilities in cellular processes and physiological pathways without interfering with the normal biological system. Nanomaterials used in drug delivery have great potential to carry large amounts and different types of biological cargo. The nanosystem abates the drug from rapid degradation and clearance through the reticuloendothelial system. The surface can be modified to react with environmental factors giving responsive drug release [75,76,77]. Different types of NPs are new forms of materials with promising biological properties and low toxicity and seem to have a high potential for passing through physiological barriers and accessing specific target tissues [78]. 

### 3.2. Metal Nanoparticles and Sperm Cryopreservation

Apoptosis, reduced cellular metabolism, and defective acrosome reaction are commonly caused by the increase of ROS levels [79]. Durfey et al. [80] used conjugated magnetic NPs for molecular-based selection of boar spermatozoa, and results showed that the nanoselected spermatozoa had improved motion characteristics with a higher proportion of progressive spermatozoa and straightness. Other reports [81,82] used NPs from FeO conjugated with annexin V to determine the early apoptosis of porcine and bovine sperm cells, respectively. 

The use of antioxidants, such as nano-zinc oxide, can be important in reducing ROS generation and increasing sperm survival [75,76,77,83,84]. Using zinc nanoparticles (50 µg/mL) or selenium nanoparticles (1 µg/mL) in a SHOTOR extender enhanced morphological characteristics and ultrastructure of camel epididymal spermatozoa after cryopreservation via the reduction of apoptosis and lipid peroxidation [60]. 

In Holstein bulls, supplementing a semen extender with Se-NPs (1.0 µg/mL) improved post-thaw sperm quality and conception rate through reducing apoptosis, LPO, and sperm damage [85]. Moreover, in rams, Hozyen et al. [86] and Nateq et al. [87] used SeNPs (1 µg/mL) and showed improvement in motility, viability index, and membrane integrity, while acrosome defects, DNA fragmentation, and malondialdehyde (MDA) concentrations were reduced. 

The addition of green synthesized gold nanoparticles (GSGNPs) (10 ppm) to a Tris-based extender improved buck semen freezing by maintaining the sperm membrane and acrosome integrity post-thawing. In addition, GSGNPs improved antioxidant capacity and consequently scavenged ROS in a buck semen extender [88]. GSGNPs are nontoxic and possess several medical applications [89]. While gold and silver NPs can penetrate the plasma membrane and can be detected inside the human sperm nucleus [90], no evidence regarding their spermatoxicity has been reported (Table 1). 

### 3.3. Herbal Extract Nanoparticles and Sperm Cryopreservation

Recently, several studies examined herbal extracts as natural antioxidants and suppressors of lipid peroxidation in semen preservation of farm animals. For instance, *Moringa oleifera* leaf extract improved the antioxidative defense for cryopreserved ram and buffalo spermatozoa [96,97]. *Arctiumlappa* root extract improved spermatozoa survivability and abnormality with appropriate progressive motility when used as a supplement with cryopreserved ram semen [98]. Curcumin extract exerted antioxidative effects and improved spermatozoa post-thaw quality when used as a supplement with cryopreserved bovine and rabbit semen [95,99,100]. Moreover, *Alnusincana* bark extract [101] and *Albiziaharveyi* leaf extract [102] showed protective antioxidative effects when used as a supplement with cryopreserved ram and bovine semen, respectively. Ginger and echinacea extracts improved the spermatozoa quality and fertilization ability when used as a supplement with cryopreserved ram semen [103]. To this end, Ismail et al. [94] reported that mint, thyme, and curcumin extract nanoformulations enhanced sperm functions and redox status of post-thawed buck semen and decreased sperm apoptosis and chromatin decondensation. Supplementing the extender with curcumin nanoparticles (1.5 µg/mL) also improved the quality of post-thawed rabbit sperm by reducing apoptosis and enhancing antioxidative defense [95] (Table 1).

### 3.4. Vitamins Nanoparticles and Sperm Cryopreservation

Vitamin E nanoemulsions (NEs) protected red deer epididymal sperm from oxidative damage, maintained mitochondrial activity, protected the acrosome integrity, prevented cell death, and reduced ROS and LPO after OS induction (with 100 μM Fe^2+^/ 500 μM ascorbate) and hence improved sperm velocity and progressive motility [104].

## 4. Artificial Exosome-Like Vesicles (Liposomes) for Semen Cryopreservation

A liposome is a spherical nanovesicle with a single lipid bilayer that is produced artificially through disrupting plasma membranes via sonication [105]. Liposomes can be used as a vehicle for delivering nutrients and drugs to target tissues [106,107]. Liposomes can be loaded with antioxidants such as lycopene [108] and quercetin [109] and result in a significant increase in sperm total and progressive motility as well as increased viability, plasma membrane integrity, and mitochondria activity in rooster spermatozoa. Moreover, liposomes can be loaded with lipid-related content (such as lecithin [110]) to improve the plasma membrane regeneration efficacy during the freeze–thaw process of ram spermatozoa. Liposomes were used as a cryoprotectant additives in several animal species including equine [111,112], buffalo [113], ovine [107,114,115], porcine [116], and bovine [117] with reported improvement in fertility after AI [118]. It has been proposed that liposomes with their contents of phospholipids (phosphatidylserine, dioleoylphosphatidylcholine, phosphatidylcholine, dipalmitoylphosphatidylcholine, and dimyristoylphosphocholine) and saturated and unsaturated fatty acids can fuse with the sperm plasma membrane and abate the damage to spermatozoa caused by the freeze–thaw process [119,120] (Figure 1). For instance, in rams, liposomes comprising egg-phosphatidylcholine and dipalmitoylphosphatidylcholine used as a supplement with washed spermatozoa provided immediate protection against cold shock as indicated by motility preservation [121]. Similarly, in stallions, liposomes comprising a mixture of egg phosphatidylcholine and phosphatidylethanolamine (named E80-liposomes) were efficient in preserving post-thaw sperm motility [112]. In contrast, in bovines, liposomes composed of dioleoyl-glycero-phosphocholine and dioleoyl-glycero-phospho-glycerol resulted in higher post-thaw survival, progressive motility, and acrosome reaction when compared to dioleoyl-glycero-phosphocholine alone [117]. The transition of lipid to gel phase during cooling and freezing is highly dependent on the lipid composition of the membranes, and therefore the liposome fusion facilitates lipid and cholesterol transfer, which leads to rearrangement of cell membrane components and modifies the membrane physicochemical properties, thereby improving the cryostability of the spermatozoa [117,118]. OptiXcell^®^ is one such commercial product that uses the liposome-based commercial extender and is currently used for several animal species [122,123,124,125].

## 5. Potential Uses of Exosomes in Semen Cryopreservation

Extracellular vesicles (EVs) including exosomes are membrane-bounded nanovesicles containing proteins, lipids, and nucleic acids (microRNAs and mRNAs) involved in cellular communication [126,127]. A wide variety of cells release EVs in physiological and pathological circumstances [128]. EVs play major roles in numerous biological communications, including reproduction, serving as potential theranostic candidates for normal and abnormal conditions [129].

Unlike other EVs, exosomes are secreted from cells by the exocytosis pathway. Exosomes are like a snapshot of the originating cells, and the variability of the secreting cell is reflected in the exosomal compositions [126]. Once these exosomes are taken by target cells, they transfer their cargo, which includes proteins [130,131], miRNA [132,133], and mRNA [134,135,136], to the target cells (Figure 1). This cargo may participate in energy pathways, protein metabolism, and maintenance of recipient cells.

Thus, exosomes confer different epigenetic and phenotypic modifications on recipient cells that affect the viability, tolerance to the external factors, and regenerative capabilities of their target cells [137]. Exosomes have also been found to play important bioactive functions such as sperm maturation, capacitation, acrosome reaction, and fertilization [138]. Recent findings regarding the regenerative potential of exosomes have guided the research towards the exploitation of exosomal potential for improving the outcomes of sperm freezing [137].

Different growth factors associated with exosomes have been reported to play an active role in the repair and accelerated healing of damaged tissue [139]. Additionally, the therapeutic potential of exosomes has also been reported to be effective in arthritis, diabetes [140], immunotherapy, nervous system-related issues [141], cellular aging, and tumors [142]. Similarly, the treatment of spermatozoa with exosomes during the freezing procedure was found effective in improving the post-thaw quality of canine [137], porcine [138], and rat semen [143]. 

### 5.1. Effect of Exosomes on Sperm Motility and Viability

Motility and viability of spermatozoa are very important quality-related parameters that have a direct influence on fertility. A strong correlation was found between increasing the concentration of seminal plasma exosomes and the sperm motility and viability of boar spermatozoa [138] when preserved at liquid stage (17 °C for 10 days). Moreover, mesenchymal stem cell (MSC)-derived microvesicles improved the frozen/thawed quality of rat spermatozoa [143]. It has been proposed that MSC-derived microvesicles shuttle surface adhesion molecules, such as CD54 (ICAM-I), CD106 (VCAM-I), CD29 (β1-Integrin), and CD44, and consequently increase the adhesive properties of sperm [143]. This improved motility was demonstrated in liquid storage (17 °C) [138] as well as frozen dog [137] and rat spermatozoa [143]. The amplitude of lateral head displacement also improved in exosome-treated dog spermatozoa [137]. Interestingly, stem-cell-derived conditioned medium and exosomes improved motility, viability, mitochondrial activity, and membrane integrity post-thawing in canine semen cryopreservation [144,145].

### 5.2. Effect of Exosomes on Sperm Capacitation and Structural Integrity

The structural integrity of spermatozoa is considered imperative for the proper functioning and fertilization of oocytes. The structures including the plasma membrane (physiological barrier [138]), acrosome (sperm penetration), and chromatin (embryo quality [146]) affect gamete interaction and embryonic development. Damage to these structures can lead to fertilization failure. Exosomes could transfer spermadhesins (AWN and porcine seminal protein, PSP-1) to the sperm membrane that could help to maintain sperm function through inhibiting premature capacitation (decapacitation) during long-term liquid storage [138]. Similarly, exosomes derived from mesenchymal stem cells increased the fraction of sperm with an intact acrosome and increased the expression of transcripts related to the repair of the plasma membrane (ANX 1, FN 1, and DYSF) and chromatin material (H3 and HMGB 1) in frozen/thawed dog spermatozoa [137]. In bovines, oviduct-derived EVs significantly stimulated the acrosome reaction by increasing the levels of protein tyrosine phosphorylation (PY) and increasing intracellular calcium levels in frozen/thawed spermatozoa [147]. In wildlife animals (red wolves and cheetahs), oviduct-derived EVs showed improvement in sperm motility and acrosome integrity and prevented the premature acrosome reaction post-thawing [148].

Capacitation is a physiological process that enables the spermatozoa to fertilize the oocytes. Naturally, capacitation occurs during spermatozoa transit through the uterus and oviduct. In vitro storage of spermatozoa requires inhibition of premature capacitation for maintaining sperm survival [138]. The higher concentration of seminal plasma isolated exosomes significantly decreased the percentage of capacitated spermatozoa upon artificially induced capacitation using 3 mg/mL BSA [138].

### 5.3. Effect of Exosomes on Antioxidant Capacity

Oxidative stress is one of the major causes of low fertility of post-thaw spermatozoa [149]. A. Mokarizadeh et al. [143] reported increased antioxidant activity in frozen/thawed rat spermatozoa treated with exosomes during freezing, i.e., decreased expression of mitochondrial ROS modulator (*ROMO1*) gene in exosome-treated spermatozoa [137]. Moreover, Du et al. [138] showed increased total antioxidant capacity activity and decreased malondialdehyde content when diluents were supplemented with seminal plasma exosomes. It was hypothesized that the enhanced antioxidant capacity of spermatozoa was either due to the horizontally transferred antioxidant and other factors including mRNA and proteins from exosomes or due to the modified hydrophobic character of the membrane. Table 2 describes the available literature that used exosomes for semen preservation either in cooling or in freezing.

## 6. Conclusions

Current trends include using nanoparticles and natural or artificial nanovesicles such as exosomes and liposomes to improve the cryopreservation of semen. Nanoparticles mostly work as antioxidants (Figure 2) with significant effects when compared with corresponding metals or herb extracts. The functional molecules present inside the exosomes such as miRNA, mRNA, and proteins (Figure 2) are involved in the proper execution of a wide variety of physiological interactions that can help resolve issues related to the fertility of male gametes. Liposomes, with their contents of phospholipids and lipid chains, can replace the damaged lipid skeletons of the frozen/thawed spermatozoa. The treatment of spermatozoa with exosomes improved the efficiency of freezing procedures; however, further in vivo and fertility studies are essential to investigating the influence of exosome treatment on sperm functions. Since liposomes are currently available as a commercial product for semen cryopreservation, nanoparticles and nanoformulations as well as EVs and exosomes derived from the reproductive tract or stem cells should adhere to the appropriate manufacturing practices, quality control measurements, and safety and efficacy protocols for commercial purposes in AI.

## Figures and Tables

**Figure 1 animals-10-02281-f001:**
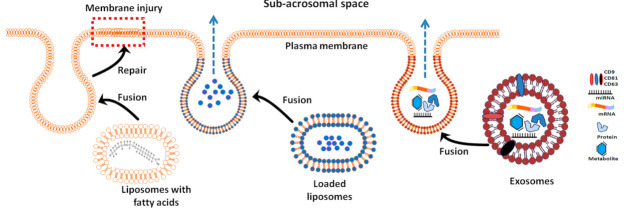
The proposed mechanism of spermatozoa protection through exosomes and liposomes. Liposomes with their contents of fatty acid can replenish the damaged sperm plasma membrane caused by freezing/thawing. Liposomes when artificially loaded with certain chemicals and exosomes with their contents of miRNA, mRNA, proteins, and metabolites can fuse and transfer their cargo into the subacrosomal space and inside the spermatozoa.

**Figure 2 animals-10-02281-f002:**
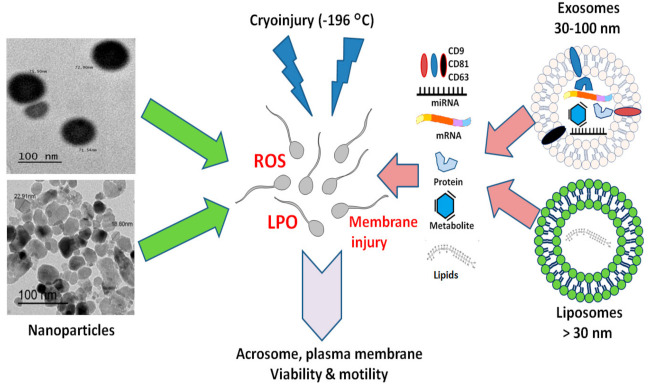
The overall effects of nanoparticles, exosomes, and liposomes in improving semen cryopreservation and reducing cryoinjury. Nanoparticles either from metals or from natural herbs act mainly as antioxidants, while exosomes can deliver bioactive components such as antioxidant enzymes, proteins, lipids, mRNA, and miRNA to protect sperm against cryoinjury such as that caused by reactive oxygen species (ROS) and lipid peroxidation (LPO). Liposomes can fuse with the sperm plasma membrane and replenish the damaged phospholipids caused by freezing/thawing.

**Table 1 animals-10-02281-t001:** Summary of the current reports using nanoparticles (NPs) for semen cryopreservation.

Animal Species	Nanoparticle (NPs)	The Effects	Reference
Goat bucks	Nano-lecithin	Improved motility, viability, and hypo-osmotic swelling test and lower apoptosis.	[91]
Bulls	Nano-lecithin-based extender with glutathione peroxidase	Enhanced plasma membrane integrity and reduced malondialdehyde (MDA) concentration.	[92]
Bulls	Selenium NPs	Improved kinematic sperm quality, antioxidative defense, and decreased apoptotic and necrotic cells.	[85]
	Zinc NPs	Improved plasma membrane integrity and mitochondrial functions.	[93]
Camel	Selenium NPs Zinc NPs	Improved sperm functions (progressive motility, vitality, sperm membrane integrity). Maintained ultrastructural morphology and decreased apoptosis. Increased antioxidative defense.	[60]
Goat	Mint, thyme, and curcumin nanoformulations (NFs)	Improved progressive motility, vitality, and plasma membrane integrity; antioxidative defense; chromatin decondensation. Decreased apoptosis.	[94]
Goat	Green synthesized gold nanoparticles (GSGNPs)	Improved motility, survivability, membrane integrity, acrosome integrity, and antioxidative defense.	[88]
Rabbit	Curcumin NPs	Enhanced sperm motility and antioxidative defense. Reduced apoptotic and necrotic spermatozoa.	[95]

**Table 2 animals-10-02281-t002:** Main literature reporting the beneficial effects obtained following the supplementation of exosomes for semen preservation.

Species	Sources of EVs/Exosomes	Condition of Storage	The Improved Parameters	References
Pig	Seminal plasma	17 °C for 10 days	Viability, motility, plasma membrane integrity, antioxidant capacity, and MDA reduction	[138]
Pig	Oviduct-derived	Freezing	Survival and motility	[150]
Rat	Bone marrow-derived mesenchymal stem cells	Freezing	Viability, motility, total antioxidant capacity, and increased surface adhesion molecules	[143]
Dog	Amniotic-derived mesenchymal stem cells and conditioned medium	Cooling and freezing	Viability, motility	[144,145]
Dog	Adipose-derived mesenchymal stem cells	Freezing	Viability, motility	[137]
Red wolves and cheetahs	Oviduct-derived	Freezing	Motility and acrosome integrity	[148]
Bovine	Oviduct-derived	Freezing	Viability	[147]

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
