# Peer review of "The Current Trends in Using Nanoparticles, Liposomes, and Exosomes for Semen Cryopreservation"

_animals, 2020, doi:10.3390/ani10122281_

Round 1

Reviewer 1 Report

The AA revised the manuscript according to the referee's suggestions. The paper is now suitable for publication.

Reviewer 2 Report

Dear authors, 

I would like to report to you that the revised manuscript  that has been adjusted following the indications of the referee, could be adapt for publication in my opinion. 

regards 

Reviewer 3 Report

The authors made all the changes suggested by the referee

This manuscript is a resubmission of an earlier submission. The following is a list of the peer review reports and author responses from that submission.

Round 1

Reviewer 1 Report

This manuscript aimed to review some of the current strategies of using nanoparticles and nanovesicles (liposomes and exosomes) to combat the cryoinjuries associated with semen cryopreservation. 

After having introduced the detrimental effects connected to cryopreservation AA focused their attention on the use of nanoparticles, liposomes, and exosomes as supplements.

 The topic can be of interest to the reader of Animals; however, in the referee's opinion, a more deep and detailed analysis of these new methods could be more attractive. AA often cited Artificial Insemination, routinely used in zootechnical animals,  but the paper is mainly focused on reviewing recent literature on sperm cryopreservation, one of the Assisted Reproductive Technologies. Undoubtedly, cryopreservation exerts evident alterations on sperm mainly, as here explained, by oxidative stress. However, a more detailed presentation of these new methods, also with a practical approach probably useful for nonexperts, would be more interesting.  

I warmly encourage AA in improving this review, its organization, and its depth, also by adding and appropriately commenting on recent literature not cited here.

Reviewer 2 Report

The manuscript entitled “The current trends in using nanoparticles, liposomes, and exosomes for semen cryopreservation”  results as interesting  because it deals with an innovative topic namely the use of NPS in semen cryopreservation.

However, I feel sorry to say that your  paper needs major revision before publication for the following reasons:

  • Although the referee isn’t a mother tongue the manuscript results difficult to read because of grammar errors. In addition, please check the format of the text and font size.
  • In introduction: it should be explained better what the nanotechnologies in the reproduction animal field are and specify what nanotechnologies the author wants to deal with in the review
  • Connect AI with cryopreservation better. You could write something similar to this: The semen cryopreservation contributes to genetic improvement through artificial insemination, eliminates geographical barriers in artificial insemination application and supports the preservation of endangered breeds thus the conservation of biodiversity. But the sperm freezing process induces ultrastructural, biochemical and functional changes of spermatozoa. Especially, spermatozoa’s membranes and chromatin can be damaged, sperm membranes’ permeability is increased, hyper oxidation and formation of reactive oxygen species takes place, affecting fertilizing ability and subsequent early embryonic development.

paragraph 4.3 (Herbal extract nanoparticles and sperm cryopreservation):The referee suggests to  add more information specifying the animal species where beneficial effects were obtained

In table 1: the title results as not clear, please change for Ex: “Main  literature reporting the use of NPs in the semen cryopreservation process”. In addition, the authors must specify if the use of NPs is only when frozen semen is used or also when it is stored at 4°C (see Cerium oxide in table 1)

Please add more information in table 1 about the effects of cerium oxide.

Lines 225 to 230 Add references and specify the animal species 

Table 2

  • change the title (see revision table 1). Es Main literature reporting the beneficial effects obtained following the supplementation of …..
  • In table 2, first row, improved parameters is it Livability or viability ? please, check!

Reviewer 3 Report

The article of Saadeldin et al. corresponds to a bibliographic review focused on the effects of the addition of new cryoprotecters as nanoparticles (NPs), exosomes, and liposomes in the sperm freezing media of different species (bovines, pigs, mice, rabbits, camel, etc).

The topic is interesting and novel, providing recent information regarding new strategies to improve semen conservation protocols.

Major Comments:

  1. a) The review not only refers to cryopreservation, but also to refrigeration at 17 or 5 ° C, and throughout the manuscript, it is not always clear whether the cited studies use frozen-thawed or refrigerated semen.
  2. b) The manuscript extensively addresses the concept of oxidative stress describes seminal plasma and antioxidants and their role in sperm function, also continuously associates the positive effects with a reduction in ROS levels, therefore, the title should change to "cryopreservation" by “preservation” and include “and its effects on oxidative stress”.
  3. c) In sections 2 and 3, it is not clear whether the harmful effect of free radicals is the same for sperm from different animals mentioned, considering that the composition of the membrane differs between species, in the same way, the species and in which medium, also in which NPs, exosomes, and liposomes were used throughout the manuscript.

Minor comments:

Line 70-71: indicate the species

Line 108-113: indicate the species

115-130: Provide background information on the effect of seminal plasma in refrigeration, freezing, and/or thawing media, although the title of this paragraph says the antioxidant system of seminal plasma, the information provided points only to two antioxidants that under natural conditions are in seminal plasma, are there not others? Is this composition the same between species? Section 2 and 3 could be merged. "Seminal plasma and antioxidants, their effect on sperm function"

Line 165-167: indicate the species, in which medium the NPs were added and the effect on sperm function of studies 80, 81, and 82.

Line 186-189: Indicate as many times as necessary which species is being referred to.

The description in table 1 indicates NPs for cryopreservation, but in the first file of the table it says preservation, I recommend deleting references 103-105 from the table (they could be included in section 4.2 as antecedents in refrigeration), considering that they are studies in refrigeration.

Line 181-184: in sperm? Indicate species.

Line 197: indicate how OS was induced

Line 226-230: species and citations need not be included, apparently, they are different studies, since in one there was a positive effect on motility and in the other, there was no effect.

Line 241-243: indicate the preservation method in which the prior mentioned studies were carried out.

Table 2: change cryopreservation for preservation.

Line 270-275: indicate the species.

Line 281: check the spacing, this is repeated throughout the manuscript, always with a reference.